# Alzheimer’s Disease, Sleep Disordered Breathing, and Microglia: Puzzling out a Common Link

**DOI:** 10.3390/cells10112907

**Published:** 2021-10-27

**Authors:** Tyler K. Ulland, Andrea C. Ewald, Andrew O. Knutson, Kaitlyn M. Marino, Stephanie M. C. Smith, Jyoti J. Watters

**Affiliations:** 1Department of Pathology and Laboratory Medicine, University of Wisconsin Madison, Madison, WI 53705, USA; tulland@wisc.edu (T.K.U.); kmmarino@wisc.edu (K.M.M.); 2Neuroscience Training Program, University of Wisconsin Madison, Madison, WI 53705, USA; 3Department of Comparative Biosciences, University of Wisconsin Madison, Madison, WI 53706, USA; aewald@wisc.edu (A.C.E.); aoknutson@wisc.edu (A.O.K.); stephanie.m.smith@medtronic.com (S.M.C.S.)

**Keywords:** neuroinflammation, intermittent hypoxia, inflammasome, animal models, sexual dimorphism

## Abstract

Sleep Disordered Breathing (SDB) and Alzheimer’s Disease (AD) are strongly associated clinically, but it is unknown if they are mechanistically associated. Here, we review data covering both the cellular and molecular responses in SDB and AD with an emphasis on the overlapping neuroimmune responses in both diseases. We extensively discuss the use of animal models of both diseases and their relative utilities in modeling human disease. Data presented here from mice exposed to intermittent hypoxia indicate that microglia become more activated following exposure to hypoxia. This also supports the idea that intermittent hypoxia can activate the neuroimmune system in a manner like that seen in AD. Finally, we highlight similarities in the cellular and neuroimmune responses between SDB and AD and propose that these similarities may lead to a pathological synergy between SDB and AD.

## 1. Introduction

Although neurodegenerative diseases like Alzheimer’s Disease (AD), are strongly associated with sleep disordered breathing (SDB), the cellular and physiologic mechanisms underlying this relationship remain poorly understood. Thus, the goals of this review are to: (1) briefly summarize current knowledge about SDB and AD to call attention to gaps in knowledge, (2) present experimental evidence supporting the hypothesis that the neuroimmune system is a common point of convergence in both SDB and AD pathologies, and (3) highlight the paucity of understanding of basic cellular mechanisms underlying the intersection between SDB and pathology in AD, with a focus on the neuroimmune response.

## 2. Neurodegeneration Is Common in Both SDB and AD

### 2.1. Sleep Disordered Breathing

Sleep disordered breathing (SDB) occurs when individuals repetitively stop breathing during sleep, often hundreds of times each night. There are three major types of SDB: (1) obstructive sleep apnea (OSA) where the upper airway either partially or fully collapses during inspiration, (2) central sleep apnea where respiratory rhythm generating neurons in the brainstem fail to send sufficient signals to upper airway pharyngeal dilator muscles and chest wall respiratory pump muscles, and (3) mixed or complex sleep apnea, which involves a combination of both obstructive and central apneas. OSA is by far the most common form of SDB, the prevalence of which is increased in individuals with larger neck circumferences, often due to adipose tissue deposition around the neck that creates anatomical impediments on the airway [1,2,3,4,5]. Sleep apnea is estimated to affect ~17–26% of the adult population in the US, with up to 80% of people with the disorder remaining undiagnosed [6,7,8]. The prevalence in the United States of mild, moderate, and severe OSA is 10%, 3.8% and 6.5% respectively [9], with the remainder of individuals effected by sleep apnea having central or mixed apneas. It is thought that some obstructive events may begin as a central event first [10,11]. Cessations in breathing due either to central or obstructive apneas can cause intermittent hypoxia (IH). IH results when breathing resumes between episodes of recurrent apneas or hypopneas, reductions in breathing, causing intermittent re-oxygenation of blood and tissues [12,13]. IH is thought to be causal of several morbidities associated with SDB including cardiovascular remodeling, metabolic disorder, sympathetic nervous system hyperactivation, and cognitive impairments [9].

### 2.2. Neurodegeneration in SDB

Cognitive impairments in individuals with SDB present as attention deficits, memory loss, and decreased executive functioning [14]. These cognitive impairments correlate with neuroanatomical changes and gray matter loss, based on magnetic resonance imaging morphometry [15]. The central nervous system (CNS) regions most affected by SDB are associated with memory, attention, and higher cognitive function and include the hippocampus; frontal, parietal, and temporal cortices; and brainstem regions including the nucleus tractus solitaries [16]. There is also gray matter loss in cerebellar, motor area, and limbic regions associated with upper airway motor regulation and cognitive processing [17]. Accordingly, exposure of rodents to IH recapitulates some but not all aspects of neuronal loss observed in human SDB. A seminal paper by Gozal et al. demonstrated cortical and hippocampal neuronal apoptosis in rats exposed to IH; this neuronal loss correlated with spatial learning deficits [18]. As the first study to report neurocognitive deficits in an animal model of IH, it sparked intense research efforts into the neuropathological consequences and molecular mechanisms underlying IH-induced CNS injury leading to similar observations in mice and other rat SDB paradigms [19,20,21,22,23,24,25]. While rodent IH models do not display neuronal apoptosis in all CNS regions observed in human SDB patients, this could be due to the short durations of experimental IH exposures (usually only weeks), and/or the absence of other morbidities of SDB such as sleep fragmentation and intermittent hypercapnia that typify human SDB. Sleep fragmentation refers to the overall amount and distribution of wakefulness during a total sleep period.

SDB is associated with neurodegenerative diseases; over 50% of patients with AD experience SDB [26]. Despite the prevalence, the impact of SDB on the progression of neurodegenerative and genetic disorders is less well understood. A study in a small population of AD patients with OSA found that continuous positive airway pressure improved cognitive scores in the fields of memory and attention [27], suggesting that some of the cognitive deficits exhibited in these patients could be slowed or even reversed by treating the OSA aspect. Although these studies need to be repeated in larger patient cohorts, evidence from animal IH studies support this general notion. For example, IH exposure increases cerebral and hippocampal amyloid beta (Aβ) burden [28,29,30] and hypoxia induces tau hyperphosphorylation [30,31,32], two pathologic hallmarks of AD that will be discussed further below.

### 2.3. Experimental Models of SDB

Rodent models mimicking key SDB aspects, most commonly IH, have been developed to better understand mechanisms of SDB [33,34,35,36]. Importantly, IH alone can recapitulate most of the OSA-associated morbidities observed in humans [37], thus disentangling the complexities of OSA in animal models. The hypoxia exposure paradigms used in rodents most often range between 5% and 11% O2, while the cycle frequency, cycle length, and daily duration depend on whether rats or mice are used. Because mice are efficient at physiologically adapting to hypoxia, inducing pathologic IH requires lower O2 levels, more episodes per hour, and longer total exposure times than rats [18,38,39,40]. Sleep fragmentation, a common IH co-morbidity [41,42], can also be modeled in rodents using mechanical devices that interrupt sleep without causing sleep deprivation or reduction in total sleep [43,44,45,46,47,48]. Sleep fragmentation alone, like IH, can also recapitulate some negative aspects of SDB including hypertension and metabolic disturbances [49,50]. Recent studies have begun to tease apart the individual physiologic effects of IH and sleep fragmentation [51].

### 2.4. Alzheimer’s Disease

AD is the most prevalent form of dementia among the elderly. As many as 5 million Americans, about 1 in 10 people over the age of 65, have AD [52]. In the United States, AD is ranked as the 6th leading cause of death [53]. Most observational and experimental evidence to date indicates that AD is initiated by the accumulation of extracellular plaques composed of Aβ leading to progressive accumulation and aggregation of hyperphosphorylated tau into neurofibrillary tangles (NFTs) [54]. These plaques and NFTs then contribute to neuronal loss, brain atrophy, and ventricular enlargement [55,56]. Clinically, AD is characterized by progressive loss of memory, emotional and cognitive changes, and diminished reaction to stimuli [57]; typically, onset of noticeable symptoms occurs after 60 years of age. Although family history, genetics, environment, lifestyle and SDB are all likely contributing factors to disease development and progression, there is no definitive predictive factor for AD [30,57].

Of the various hypotheses on the etiology of AD pathogenesis, the foremost three are the: (1) amyloid, (2) tau, and (3) infectious origin hypotheses. The *amyloid hypothesis* states that the deposition of extracellular plaques, which are primarily composed of Aβ peptides, is the causative agent underlying AD pathology [58]. Although the *amyloid hypothesis* is the most widely accepted, it does not explain all aspects of AD pathogenesis [59,60]. The second major hypothesis is the *tau hypothesis*. This hypothesis posits that tau hyperphosphorylation and accumulation into intraneuronal NFTs is the causative agent of AD [61]. The *tau hypothesis* can explain questions and gaps left by the amyloid hypothesis, but it fails explain the full progression of AD pathology on its own. The final prominent hypothesis for AD pathogenesis is the *infectious origin model*. This hypothesis proposes that the initiating step in AD development is a pathogenic insult, either viral or bacterial, that triggers the formation of amyloid plaques and NFTs resulting in the onset of AD [62]. The *infectious origin hypothesis* is relatively new and much of the data supporting it remains circumstantial.

### 2.5. Experimental Models of AD

Several animal models of AD have been developed and have significantly contributed to our understanding of AD pathogenesis, foremost among which is the mouse. Due to the genetic manipulability of mice, several models with altered tau, such as the P301S line PS19, Aβ models including the 5XFAD and APPPS1, or mice expressing both altered tau and Aβ such as the 3xTg mice [63,64,65,66] have been developed to elucidate individual aspects of AD pathology including the formation of NFT or Aβ plaques. More complex models combine both Aβ and NFT aspects; for example, using an Aβ mouse model as a base, then seeding the brain with NFTs to explore their combined effects on AD pathology [67]. Many of these models also exhibit behavioral defects alongside cellular responses, particularly microgliosis and microglial activation, which closely mimic those observed in humans with AD. In addition to mouse models, several non-human primate and other rodent models exist. However, the use of these models has been much more limited [68].

## 3. Neuroinflammation in AD and SDB

### 3.1. Microglia and the Inflammasome in AD and SDB

One of the primary functions of microglia is to identify and resolve oxidative stress and inflammation in the CNS. Novel imaging techniques (including non-invasive two-photon in vivo microscopy) have shown the normal state of microglia to be anything but “quiescent” or “resting,” as they are highly dynamic cells; constantly surveying the brain parenchyma with their processes to interact with neurons, synapses, axons, other glial cells, and blood vessels [69,70,71,72,73]. Microglia are involved in synaptic plasticity, neurogenesis, learning and memory and functional brain connectivity via their synaptic pruning activities [74,75]. Since neuroinflammation occurs in both AD and SDB, microglia have become a recent focus of research, separately, in the AD and SDB fields.

While it has been known for more than a century that microglia are both activated and proliferative in AD, their contribution to AD pathology remains incompletely understood [76]. The identification of several single nucleotide polymorphisms in microglial genes associated with increased risk for developing AD [77,78,79] prompted extensive follow-up work that has firmly implicated microglia as key players in AD etiology [76].

Microglial function in AD is complex, having both detrimental [80,81] and neuroprotective activities [82,83,84,85]. In early AD, microglia in Aβ models work to clear amyloid plaques but also release pro-inflammatory cytokines and produce apoptosis-associated speck-like protein containing a CARD (ASC specks). ASC specks in microglia are hallmarks of inflammasome activation and seed new Aβ plaque formation [80,81,83,84,85]. In 5XFAD mice, inhibiting this inflammasome activity early on can decrease pathology [86]. Depletion of microglia from 5XFAD mice by pharmacological inhibition of the colony stimulating factor 1 receptor with PLX3397 improved hippocampal-dependent contextual memory and lessened neuronal loss, but it did not alter plaque load [87,88]. However, these findings are complicated by the fact that most of the remaining (non-depleted) microglia were heavily associated with Aβ plaques [88,89]. Therefore, the implications of these studies on determining the role of microglia in AD remain unclear. In tau mouse AD models, microglia enhance neurodegeneration [90]. However, in more complex AD models in which both Aβ and NFTs are present in a single animal, microglia appear to surround Aβ plaques and limit their neurotoxic impact resulting in reduced formation of NFTs [67]. Taken together, these data suggest a complicated role for microglia in the response to AD pathology.

### 3.2. Astrocytes in AD and SDB

Astrocytes are a population of cells in the CNS that normally regulate the integrity of the blood brain barrier and sustain normal tissue and cellular function by maintaining neuronal cell function and regulating tissue homeostasis. Although less studied, astrocytes also play a role in AD. They proliferate, lose their exclusive spatial autonomy and are associated with both tau NFTs and Aβ plaques [91,92,93,94,95,96,97]. Reactive astrocytes also produce Aβ and inflammatory cytokines that, together with those produced by microglia, contribute to enhanced neuroinflammation in the AD brain [93]. Reactive astrocytes surround amyloid plaques [98] and the magnitude of astrogliosis may correlate with plaque number [99,100,101,102]. However, it has also been reported that astrocytes associate with tau and NFTs, and reactive astrocyte response more closely correlates with NFT burden than plaque burden [91,97]. Some astrocytic processes become part of the insoluble Aβ plaque that they surround, and their other hypertrophic processes make abnormal contacts with nearby blood vessels and neurons [99]. However, astrocytes that are further away from the amyloid plaques undergo atrophy [101,103,104]. Because astrocytes become differentially activated or atrophied based on their proximity to the Aβ plaques [104,105,106,107,108], potentially prior to plaque formation, astrocytes may also contribute to AD pathology.

While considerably less is known about astrocytes in the context of SDB, a recent study of individuals with OSA showed that they had elevated levels of S100B in serum, a marker of astrocyte activation [109]. Patients with OSA also exhibit increased levels of midbrain tissue glutamate as detected by 2D magnetic resonance spectroscopy, implicating excitotoxicity and astrocyte activation [110]. Exposure of mice to IH also causes astrogliosis [18,111], but to date, there have been no studies investigating the contributions of astrocytes to IH or AD-induced neuropathology. In the one recent study currently available, in an AD mouse model (APP-PS1) IH exposure did not affect Aβ levels or plaque load, but it did increase astrogliosis (based on glial-fibrillary acidic protein; GFAP) staining [111].

### 3.3. Molecular Mechanisms of Neuroinflammation in AD and SDB

Chronic inflammasome activation is common in early onset AD patients [80] with NOD-, LRR-, and pyrin containing 3 (NLRP3) inflammasomes contributing to Aβ plaque and tau pathology in the context of Alzheimer’s Disease [80,81,112,113]. In primary microglial cultures, exposure to soluble Aβ oligomers and protofibrils results in inflammasome activation [114]. APP/PS1/NLRP3 and APP/PS1/Caspase-1 deficient mice were protected from spatial memory deficits despite having no significant difference in total brain IL-1β [80]. Although loss of NLRP3 inflammasome activation is associated with reduced tau hyperphosphorylation and improved spatial memory in mice [112], intrahippocampal injections of ASC-specks themselves resulted in the spreading Aβ pathology in double mutant APP-PSEN1 mice [81], underscoring the complexity of microglia in AD. β-hydroxybutyrate (BHB) is a byproduct of ketogenesis that is known to inhibit the NLRP3 inflammasome [115]. BHB levels are significantly lower in AD patients [86]. Supplementing 5XFAD mice with BHB resulted in significantly less plaques and less complex microglia when compared to untreated 5XFADs [86]. Along with fewer ASC specks and lowered levels of Caspase-1 in the cortex, this suggests that BHB decreases microgliosis and is an inhibitor of the inflammasome [86].

Although microglial activation of the inflammasome can cause inflammation in the brain, to date no studies have examined the impact of the inflammasome in SDB. However, one recent study started to tease apart the relationship between IH and the inflammasome. This group utilized bone-marrow derived macrophages from lean and obese mice and subsequently exposed them to IH in vitro [116]. The IH exposure increased the inflammatory profile of both sets of bone-marrow derived macrophages [116]. Pretreatment with a toll-like receptor 4 (TLR4) inhibitor also prevented IH-induced inflammation [116]. In bone-marrow derived macrophages from NLRP3-/- mice, there was no increase in the inflammatory profile due to IH exposure [116]. This study starts to uncover the role that TLR4/NLRP3 signaling roles play in exposure to IH. Our group has also examined the effects of IH on microglial inflammatory and TLR4 gene expression levels in microglia in mice in vivo, following exposure to IH for up to 14 days [117]. We found that microglial inflammatory gene expression was differentially increased in immunomagnetically isolated microglia in a region-dependent manner, and that TLR4 gene expression was similarly increased [117]. Surface TLR4 protein levels of microglia, as detected by flow cytometry, trended towards being increased at seven days of IH exposure and were significantly increased by 14 days, an effect that persisted until at least four weeks of IH exposure (Figure 1). Similarly, microglial surface CD45 protein levels were significantly increased within seven days of IH exposure, persisting through to 28 days (Figure 2). We also found that protein expression of neuronal caspase 3 was increased within seven days of IH exposure and persisted for the 28-day experimental duration (Figure 3), indicating that microglial activation and neuronal apoptosis are temporally associated. These data support the idea that microglia become activated in response to IH in vivo, and that microglial TLR4 signaling and inflammasome activation may be enhanced.

## 4. Common Mechanisms of SDB and AD

### 4.1. Bidirectional Effects of SDB and AD on One Another

In the CNS, many of the cellular processes, including the activation of microglia, are shared between SDB and AD. Only recently has the association between these processes in SDB and AD been investigated. Evidence suggests that individuals with SDB are at a higher risk for also having some form of neurodegenerative disease [12,118,119,120,121,122]. When examining institutionalized patients with dementia, it was determined that greater than 70% of those patients also had SDB, with over half of them having moderate or severe SDB [123]. This is important because this rate is nearly three times that expected in the general population, suggesting that there may be an interaction between SDB and dementia. Another study that examined post-mortem human brains (from cognitively normal individuals) found that while SDB severity was a strong predictor of Aβ plaque load in the hippocampus, it was not correlated with hippocampal NFT burden, nor Aβ plaque or NFT burden in the brainstem, where respiratory neural control resides [30]. Thus, SDB may contribute to large Aβ plaque loads, at least in pre-clinical stages of AD progression. Intriguingly, although SDB patients were also recently found to exhibit reduced CSF soluble Aβ [124,125,126,127] and increased tau proteins in CSF [125] and plasma [128,129], as well as increased Aβ plaques by PET scan [127,130,131,132,133], not all studies are consistent [127]. For example, decreased Aβ 1-42 and increased tau was reported in CSF from individuals with OSA, which also correlated with increased memory impairment [125]. However, unlike with OSA, the frequency of central sleep apnea in AD is less well understood, and the association with AD is thought to be less common than with OSA [134]. Due to the distinct cellular and pathological similarities between SDB and AD, we propose that these seemingly unrelated diseases may interact synergistically to enhance neurodegeneration and disease pathology. We also suggest that SDB-induced neurodegeneration precedes AD symptom onset, effectively setting the stage for exacerbated AD progression. In the sections below, we will describe the evidence including sleep disturbances, sexual dimorphism, common risk factors, and inflammation–particularly involving the neuroimmune system–that support the idea that SDB and AD may have common etiologies.

### 4.2. Sleep Disturbances

Unsurprisingly, SDB increases sleep disturbances and worsens the quality of sleep that is experienced, leading to daytime sleepiness, which is associated with deficits in working memory and vigilance [119]. Interestingly, sleep disturbances including insomnia, circadian rhythm alterations, and nocturnal agitation are also among the earliest behavioral changes associated with AD development, and they typically worsen during the development of the disease [121,122,134,135,136,137,138,139]. People reporting short sleep durations (<6 h) have increased incidence of hypertension, obesity, type II diabetes, cardiovascular disease, and stroke [140,141,142,143]. Sleep studies have shown that AD patients with cognitive deficits are more likely to have significantly increased sleep disturbances than AD patients with mild or no cognitive impairments, demonstrating that as AD-associated dementia severity increases, so too does the number of sleep disturbances [8]. Experimental evidence from both human studies and animal models have further highlighted the importance of sleep in AD progression, as the clearance of Aβ occurs mostly during sleep and as little as one night of disrupted sleep can lead to an increase in detectable Aβ levels in the brain [144,145]. Additionally, CSF tau, an important AD biomarker, increases following sleep deprivation in humans and in a mouse model of chronic sleep deprivation tau spreading was increased following sleep deprivation [146]. In the future, additional studies taking advantage of new technologies and approaches such as sleep monitoring by smart watches will allow us to further refine our understanding of the link between sleep disturbances and the development of AD.

Given that sleep disturbances are common in AD, and that SDB is associated with exacerbated AD related pathological changes, we suggest that SDB enhances and/or accelerates AD pathology. Patients with severe SDB exhibit hippocampal atrophy, as is observed in AD [30,147,148,149], and rats with reduced REM sleep showed reduced neuronal arborization in the hippocampus [150] and increased neuronal apoptosis in the locus coeruleus, laterodorsal tegmentum, pedunculopontine tegmentum, and medial preoptic area [140,151]. Rodents exposed to chronic sleep deprivation have decreased volume in brainstem respiratory nuclei (NTS and parabrachial nucleus) [152], in the medial prefrontal cortex [153], in the dorsal and CA2/CA3 hippocampal regions [154], and in the CA1 and dentate gyrus hippocampal regions [155], indicating a detrimental effect on key brain regions involved in cognition and autonomic nervous system function. 5XFAD mice exhibit increased sleep fragmentation [156].

### 4.3. Sex Hormones and Gender Differences

Interestingly, both SDB and AD have prominent sexual dimorphism. For SDB in general, men are two to three times more likely to be diagnosed than women, with diagnoses increasing in both sexes with age [6,7,157,158,159]. In the United States, 17–24% of men but only 9% of women aged 50–70 years old experience sleep apnea [6,7]. Men have both increasing incidence and severity of SDB as they age [7,157]. Regardless of age, increased SDB severity in men can be predicted by increased obesity, body mass index, waist-hip ratio, and neck circumference [7,157,160,161]. Many male patients who have SDB and are obese have high body mass index or waist-hip ratio, or larger neck circumference can decrease the severity of SDB by weight loss [162]. Women, unlike males, exhibit no positive correlation between obesity (body mass index) and OSA; OSA severity is not correlated to waist-hip ratio or neck circumference, nor is SDB affected by weight loss [157,162]. Women tend to be older than men at their initial diagnosis of OSA, and present with less severe disease [162].

Women with poly-cystic ovarian syndrome (PCOS), typically pre-menopausal with higher testosterone levels, tend to develop OSA at higher rates than women without PCOS, which suggests that sex hormones may play a role in OSA development [163]. Additionally, women with PCOS and men have similar patterns of adipose tissue deposition and size of airway structures [164,165]. Although the incidence of OSA does not change in premenopausal women, it increases during and after menopause [7,158,162,166,167]. However, despite this increased incidence of OSA in post-menopausal women, OSA severity does not worsen, nor does it become associated with obesity, like it is in men [7,158,162,166,167], suggesting sexually dimorphic mechanisms in SDB etiology. The increase in OSA incidence in post-menopausal women may be due to the reductions in circulating estrogen levels (and consequent loss of its neuroprotective effects) and would be consistent with the increased incidence of SDB in younger women with PCOS in which there are higher levels of testosterone, and relatively lower levels of estrogen.

Both OSA and AD are sexually dimorphic, with OSA affecting more men than premenopausal women, and AD being more prevalent in women than men [168]. The primary risk factor for developing AD is age [169]. In men, the risk of developing AD begins to rise in midlife and increases steadily thereafter [170]. Like SDB development and progression, women tend to be somewhat “protected” from AD until menopause, at which point OSA prevalence in women quickly approaches and surpasses that in men [170]. Interestingly, mirroring what is observed with OSA, younger women with PCOS are at an increased risk for developing AD [171]. Further, male AD patients have slower cognitive decline than women [172]. Accordingly, we hypothesize that sex hormones and/or sex differences play a pivotal role in the progression of these distinct diseases.

### 4.4. Common Risk Factors and Comorbidities for SDB and AD

Another overlapping aspect of both SDB and AD are disease-associated risk factors and comorbidities (Table 1). Clinical studies have shown that men with OSA are at an increased risk for developing comorbidities compared to women [158]. However, in both sexes, SDB leads to an increased risk for health disorders including cardiovascular and neurodegenerative disorders [158]. Additionally, both SDB and AD are often accompanied by hypertension [139,173,174]. Regarding known sex differences, men with OSA are more often diagnosed with hypertension, tend to have more severe hypertension, and are likely to have more apneas per hour compared to women with OSA [175,176]. Conversely, women with OSA tend to have milder hypertension that is unrelated to the number of apneas per hour [176]. This is similar to findings in rodent models of SDB where male rats exposed to IH have increased mean arterial pressures, whereas female rats do not [177]; ovariectomized females behaved similarly to males [178]. We hypothesize that since hypertension also exacerbates inflammation and oxidative stress via increased vascular dysfunction, patients with OSA and comorbid hypertension may be at increased risk of subsequently developing AD or other neurodegenerative diseases.

SDB and AD are often accompanied by increased inflammation and oxidative stress [117,139,169,173,174,179,180,181,182]. Studies in children with OSA have identified increases in circulating inflammatory markers [183] as well as the AD markers Aβ and presenilin 1 levels [184]. However, these studies were done in pediatric populations, not in aged adults who would be more likely to develop AD. Both women and men with OSA (without other comorbidities) display increased inflammatory biomarkers in blood or exhaled breath [185,186,187,188,189,190,191], as well as increased markers of oxidative stress, such as eNOS, HIF-1α, and VEGF [179]. Although many studies have measured indices of circulating systemic inflammation in individuals with OSA, few have examined CNS inflammation in humans with OSA.

Rodents exposed to IH exhibit increases in both peripheral and central inflammatory mediators as well as enhanced markers of oxidative stress [19,117,118,139,192,193,194]. IH exposure also significantly increases neuronal loss in the hippocampus and prefrontal cortex [18,19,118,194,195] indicating that repetitive hypoxic events during SDB can cause oxidative stress and inflammation in CNS regions involved in cognition. Coincidentally, oxidative stress and inflammation associated with AD are also detected in CNS regions including the hippocampus [139,196], entorhinal cortex [169,197], rostral ventrolateral medulla and solitary tract nucleus [198,199], the same regions in which OSA induces grey matter loss in the human brain (see *Neurodegeneration in SDB* section above on pg. 2). Together these data illustrate that similar areas of the brain are affected in both SDB and AD, underscoring the possibility that these diseases may act synergistically to worsen the pathology of the other.

### 4.5. Studying SDB in AD Animal Models

There are very few animal studies that have investigated the intersection of AD and OSA. One available study exposed 3xTg AD mice to IH and found increased intraneuronal Aβ amyloid production [29]. Another group looked at the effect of 4-weeks of IH on astrocytes and Aβ levels in APP-PS1 mice. They found that the Aβ levels remained unchanged, but there was a significant increase in GFAP positive astrocytes, indicative of a pro-inflammatory response [111]. This robust astrogliosis was found in several CNS regions: near blood vessels, and in the neocortex, corpus callosum, CA1 and CA2, dentate gyrus, thalamus, and striatum, an effect that was not observed as severely in the age matched WT controls [111]. Although these studies have begun to fill in the gaps related to the AD and SDB relationship, our ongoing studies aim to identify the role of microglia in the context of long term (months) IH exposure by assessing neuroimmune cell changes, microglia marker differences, and astrocyte and microglia morphologic alterations that regulate the CNS immune response.

## 5. Summary and Closing Thoughts

Here we have summarized the evidence supporting our hypothesis that there are several commonalities between AD and SDB pathology, including their co-morbid prevalence, the similar regions of brain impacted by neurodegeneration, the involvement of sleep disruption, the neuroinflammation that is driven by both inflammatory activation of microglia and aberrant astrocyte function, as well as common hormonal and sex-dependent influences. Consequently, we propose that microglial activation, via an inflammasome-dependent mechanism, bridges both disorders, possibly leading to reciprocal and synergistic exacerbation of both diseases. Although several recent studies have begun to interrogate the relationship between SDB and AD in human subjects, animal studies are needed to further probe mechanistic interactions. Additionally, data from rodent studies support the need for future experiments that explore not only the functional contributions of microglia and astrocyte activities to this interaction, but also the underlying cellular signaling pathways. Based on the AD literature, we posit that inflammasome activation will be a key point of convergence in uniting both pathologies; however, the evidence is sparse at best in the SDB literature to support this notion. Further studies are necessary in SDB models to understand the cell types that contribute to neuroinflammation and neurodegeneration, the cellular signaling pathways recruited to do so, and how those pathways intersect and link SDB and AD.

## 6. Experimental Methods

### 6.1. Intermittent Hypoxia (IH) Exposure

Adult male C57/BL6 mice were obtained from Harlan Laboratories (Madison, WI, USA) and housed in AAALAC-accredited facilities with 12 h:12 h light-dark conditions. All animal exposures were performed using a commercially designed system (BioSpherix, Redfield, NY, USA). Animals were housed in standard polycarbonate cages with access to food and water ad libitum and maintained in a specialized chamber (12 × 20 × 30 inches). Oxygen and carbon dioxide concentrations were continuously monitored by an O_2_ and CO_2_ analyzer and were changed by a computerized system controlling the gas outlets (Oxycycler model G2 and Watview software). During the sleep cycle (lights on), O_2_ concentrations were modified to generate a cyclical pattern of 6% and 21% O_2_ every 90 s (IH) [200,201]. During their wake period (lights off), O_2_ concentration was maintained at 21%. Airflow was sufficient to prevent CO_2_ accumulation, maintaining a concentration below 0.3%. Normoxic control animals were housed inside a chamber with circulating 21% O_2_ to mimic the IH exposure.

### 6.2. Tissue Isolation and Fixation for Flow Cytometry

CNS tissues were dissociated as previously described [202]. Briefly, mice were euthanized with an overdose of isoflurane and perfused intra-aortically with cold 0.1 M phosphate buffered saline (PBS). The frontal cortex and hippocampus were dissected out, placed into cold Hank’s Buffered Salt Solution (HBSS; Cellgro, Herndon, VA, USA) on ice. Tissue was mechanically dissociated and pushed through a pre-moistened 100 µM cell strainer with a syringe plunger and washed with cold HBSS supplemented with 0.01 mg/mL DNase (Worthington Biochemicals, Lakewood, NJ, USA). Dissociated tissues were resuspending in 26% Percoll (GE Healthcare, Waukesha, WI, USA) in 0.1 M PBS and centrifuged at 850× *g* for 15 min to remove myelin. Samples were fixed in a modified zinc-based fixative (mZBF) (0.5% zinc chloride, 0.5% zinc trifluroacetate, 0.05% calcium acetate in 0.1 M Tris-HCl, pH 6.4–6.7) [203] and glycerol (1:1) and stored at −20 °C overnight or until ready to be stained for flow cytometry [203,204].

### 6.3. Staining for Flow Cytometry

Fixed samples were washed 3x in ice-cold PBS. Cell surface protein stains CD11b-APC-Cy7 (1:150; eBiosciences, San Diego, CA, USA) and the active TLR4/MD2 complex (TLR4-PE-Cy7) (1:25; Biolegend, San Diego, CA, USA) were performed on ice in 50 µL of 1X PBS supplemented with 0.1% BSA for 25 min. Cells were washed 3x and resuspended in a permeabilization buffer (1X PBS + 0.2% saponin + 0.1% BSA). Intracellular staining was performed in 50 µL permeabilization buffer on ice for 45 min for NeuN-Alexa488 (1:500; Millipore, Billerica, MA, USA) and caspase-3-PE (1:20; BD Biosciences, San Jose, CA, USA). Samples were washed and resuspended in the permeabilization buffer containing DAPI (1 µg/mL; Invitrogen, Carlsbad, CA, USA) to identify cells with intact nuclei. Cells were analyzed on a BD LSR II and FACSDiva software (BD Biosciences). FCS files were analyzed via FlowJO software v10.8.0. Cells were gated to live cell singlets as previously described [205].

### 6.4. Statistical ANALYSIS

All statistical analyses were performed using GraphPad PRISM version 9.1.0. The Grubb’s Test was used to identify outliers. Statistical significance (set at *p* < 0.05) was determined using Student’s *t*-tests.

## Figures and Tables

**Figure 1 cells-10-02907-f001:**
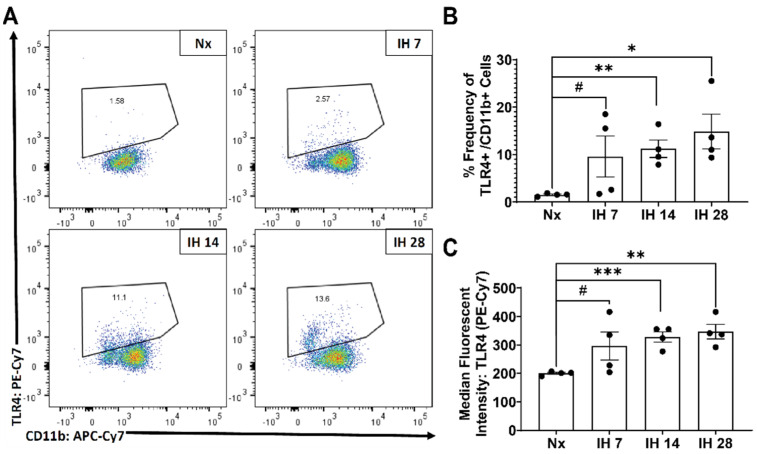
Microglial surface expression of active TLR4/MD2 is significantly increased by IH exposure in vivo. Mice were exposed to normoxia (room air; Nx), or IH for 7, 14, or 28 days. 14 h following the last hypoxic exposure, hippocampal tissue was dissociated, and single cell suspensions were stained with anti-TLR4/MD2 and anti-CD11b antibodies for analysis by flow cytometry. (**A**) Representative dot plots of TLR4/MD2 vs. CD11b immunofluorescence. (**B**) Frequency of TLR4/MD2+ cells as a percentage of total CD11b+ cells. (**C**) Average mean fluorescent intensity of TLR4/MD2 in CD11b+ cells. Statistical significance was determined by student’s *t*-tests. * *p* < 0.05; ** *p* < 0.01; *** *p* < 0.001; # 0.05 < *p*< 0.12.

**Figure 2 cells-10-02907-f002:**
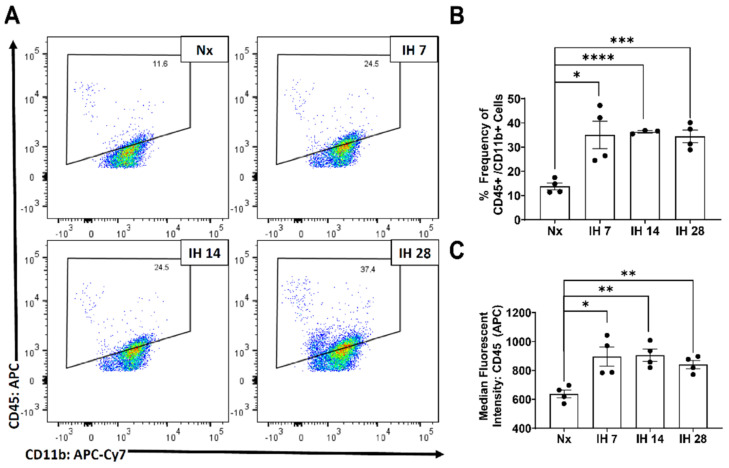
Microglial CD45 expression increases following IH exposure in vivo. Mice were exposed to normoxia (room air; Nx), or IH for 7, 14, or 28 days. 14 h following the last hypoxic exposure, hippocampal tissue was dissociated, and single cell suspensions were stained with anti-CD45 and anti-CD11b antibodies for analysis by flow cytometry. (**A**) Representative dot plots of CD45 vs. CD11b immunofluorescence. (**B**) Frequency of CD45+/CD11b+ cells. (**C**) Average mean fluorescent intensity CD45 in CD11b+ cells. Statistical significance was determined by student’s *t*-tests. * *p* < 0.05; ** *p* < 0.01; *** *p* < 0.001; **** *p* < 0.0001.

**Figure 3 cells-10-02907-f003:**
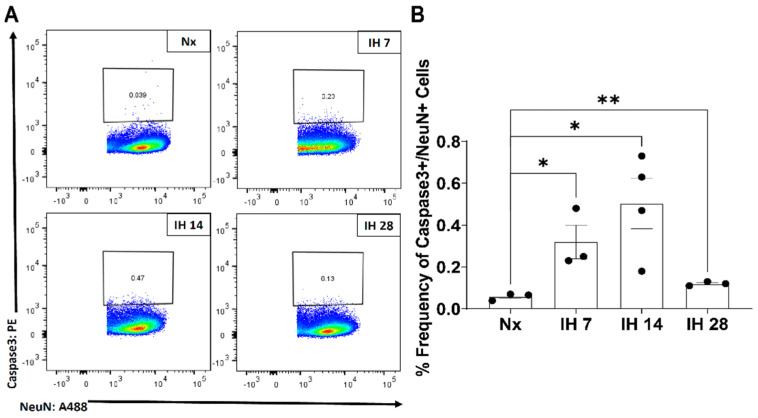
IH increases neuronal caspase-3 levels in the mouse hippocampus. Mice were exposed to normoxia (room air; Nx), or IH for 7, 14, or 28 days. 14 h following the last hypoxic exposure, hippocampal tissue was dissociated, and single cell suspensions were stained with anti-caspase 3 and anti-NeuN antibodies for analysis by flow cytometry. (**A**) Representative dot plots of caspase 3+/NeuN+ cells. (**B**) Frequency of caspase 3+/NeuN+ cells. Statistical significance was determined by student’s *t*-tests. * *p* < 0.05; ** *p* < 0.01.

**Table 1 cells-10-02907-t001:** Factors commonly observed in Alzheimer’s Disease and Sleep Disordered Breathing. “X” denotes strong experimental evidence exists; “?” denotes a potential connection.

	Alzheimer’s Disease	Sleep Disordered Breathing
Neuroinflammation	X	X
Inflammasome Activation	X	?
Hypoxia		X
Tau Pathology	X	
Amyloid β Plaques	X	?
Hypertension	X	X
Sex Differences	X	X
Neuronal Apoptosis	X	X
Microglial Activation	X	X
Astrocyte Activation	X	X
Sleep Disturbances	X	X

## Data Availability

Not applicable.

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
