# Peer review of "Alzheimer’s Disease, Sleep Disordered Breathing, and Microglia: Puzzling out a Common Link"

_cells, 2021, doi:10.3390/cells10112907_

Round 1
Reviewer 1 Report
In this review authors clearly illustrate the current knowledge about a possible link between Alzheimer’s Disease (AD) and Sleep Disordered Breathing (SDB).
The cited literature on the subject is recent and complete.
Overall the review is well written, the subject is original and interesting to the readers involved in the field of neurodegenerative diseases.
Authors introduce the reader to what is SDB and briefly describe the animal models of SDB available. Authors also give a simple and useful introduction on AD (paragraph 2.4). In this context, I would suggest to add few sentences in order to:
- Give a simple scientific definition of sleep fragmentation and sleep fragmentation index (SFI). This can be added either at line 73 or at line 94 of the manuscript.
- Add a reference on the Tau hypothesis of AD (line 119) which is missing.
In paragraph 3.3 authors include some new experiments (Figures 1-3) and the experimental description of those experiments is reported at the end of the manuscript. Since these experiments involve the use of animals, I think that the permission number for animal studies should be reported at the end of the manuscript according to the journal policy.
According to my opinion this manuscript can be accepted after performing the minor revisions described above.
Minor correction:
Line 401: “…, and effect that was observed”, should read “…, an effect that was observed”
Author Response
Point 1: In this review authors clearly illustrate the current knowledge about a possible link between Alzheimer’s Disease (AD) and Sleep Disordered Breathing (SDB). The cited literature on the subject is recent and complete. Overall the review is well written, the subject is original and interesting to the readers involved in the field of neurodegenerative diseases. Authors introduce the reader to what is SDB and briefly describe the animal models of SDB available.
We thank you for the positive comments.
Point 2: Authors also give a simple and useful introduction on AD (paragraph 2.4). In this context, I would suggest to add few sentences in order to: Give a simple scientific definition of sleep fragmentation and sleep fragmentation index (SFI). This can be added either at line 73 or at line 94 of the manuscript.
We define sleep fragmentation in line 75.
Point 3: Add a reference on the Tau hypothesis of AD (line 119) which is missing.
Thank you for pointing out our oversight - a reference is now included.
Point 4: In paragraph 3.3 authors include some new experiments (Figures 1-3) and the experimental description of those experiments is reported at the end of the manuscript. Since these experiments involve the use of animals, I think that the permission number for animal studies should be reported at the end of the manuscript according to the journal policy.
The protocol information is now included.
Point 5: Minor correction: Line 401: “…, and effect that was observed”, should read “…, an effect that was observed”
Thank you. This is now corrected.
Reviewer 2 Report
Authors reviewed the correlations between sleep disorder and Alzheimer's disease. It's a rare review on the topic but it could have significant impacts in the field. However, authors need to discuss on previous studies on the increased clearance of Abeta during sleeping and biorhythm, not only on the hypoxia. Next, authors need to discuss the studies of sleep disorder and AD with common and differential biomarkers, including active monitoring with smart watch.
Author Response
We have included discussion of this in section 4.2.